# Prevalence and Prognostic Implications of PSA Flares during Radium-223 Treatment among Men with Metastatic Castration Resistant Prostate Cancer

**DOI:** 10.3390/jcm12175604

**Published:** 2023-08-28

**Authors:** Amanjot Sidhu, Nabeeha Khan, Cameron Phillips, Juan Briones, Anil Kapoor, Pawel Zalewski, Neil E. Fleshner, Edward Chow, Urban Emmenegger

**Affiliations:** 1Sunnybrook Odette Cancer Centre, Toronto, ON M4N 3M5, Canada; amanjotsidhu7@hotmail.com (A.S.); n236khan@uwaterloo.ca (N.K.); cameron.phillips@niagarahealth.on.ca (C.P.); jbrionesc@gmail.com (J.B.); edward.chow@sunnybrook.ca (E.C.); 2Juravinski Cancer Centre, Hamilton, ON L8V 5C2, Canada; akapoor@mcmaster.ca; 3Durham Regional Cancer Centre, Oshawa, ON L1G 2B9, Canada; pzalewski@lh.ca; 4Princess Margaret Cancer Centre, Toronto, ON M5G 2M9, Canada; neil.fleshner@uhn.ca

**Keywords:** radium-223, prostate specific antigen, metastatic castration-resistant prostate cancer, Ra223-induced prostate specific antigen flare, bone metastases

## Abstract

Radium-223 (Ra233) prolongs the survival of men with symptomatic bone-predominant metastatic castration-resistant prostate cancer (mCRPC). However, prostate-specific antigen (PSA) response patterns are not closely associated with Ra223 therapy outcomes. Herein, we sought to analyze the impact of Ra223-induced PSA flares on patient outcome. Using a retrospective cohort study of Ra223 treatment in four Ontario/Canada cancer centres, we identified 134 patients grouped into sub-cohorts according to distinct PSA response patterns: (i) initial PSA flare followed by eventual PSA decline; (ii) PSA response (≥30% PSA decrease within 12 weeks of treatment); and (iii) PSA non-response. We analyzed patient characteristics and outcome measures, including overall survival (OS), using the Kaplan-Meier method and log-rank testing. PSA flares were observed in 27 (20.2%), PSA responses in 11 (8.2%), and PSA non-responses in 96 (71.6%) patients. Amongst PSA flare patients, 12 presented with post-flare PSA decreases below baseline and 15 with PSA decreases below the flare peak but above baseline. Although only six flare patients achieved ≥30% PSA decreases below baseline, the median OS of all flare patients (16.8 months, 95% CI 14.9–18.7) was comparable to that of PSA responders and non-responders (*p* = 0.349). In summary, around 20% of mCRPC patients experience Ra223-induced PSA flares, whose outcome is similar to that of men with or without PSA responses. Further studies are needed regarding suitable biochemical surrogate markers of response to Ra223.

## 1. Introduction

Prostate cancer is the second most frequent non-skin cancer in men and the cause of nearly 400,000 cancer-related deaths worldwide [1]. While localized prostate cancer is highly curable, a considerable number of patients present with either synchronous or metachronous metastatic disease, which ultimately evolves into the lethal form of prostate cancer, i.e., metastatic castration-resistant prostate cancer (mCRPC) [2]. Bone is the most common metastatic site of prostate cancer, and men with bone metastases are at risk of severe pain, pathologic fractures, spinal cord compressions, and poor quality of life [3,4,5]. However, the monitoring of bone metastases is challenging, and there is a lack of widely applicable tissue-based or circulating biomarkers [6,7,8].

Antiresorptive agents such as denosumab (a neutralizing monoclonal antibody of the receptor activator of nuclear factor kappa-Β ligand) and zoledronic acid (nitrogen-containing bisphosphonate), as well as local treatments (radiotherapy, surgery), combat bone metastases and thus reduce the frequency and/or clinical impact of skeletal-related events among men with mCRPC [5,9]. However, they do not lengthen overall survival (OS).

Radium-223 dichloride (Ra223) is a bone-seeking alpha emitter that targets areas of high bone turnover, such as bone metastases, and causes double-strand DNA breaks, thereby resulting in cancer cell death [10]. As an alpha emitter, Ra223 works at a short range, which reduces radiation exposure to healthy neighbouring tissues. While early trials documented Ra223′s low toxicity profile and analgesic properties, the phase 3 ALpharadin in SYMptomatic Prostate CAncer Patients (ALSYMPCA) trial demonstrated that Ra223 improves the OS of men with symptomatic mCRPC to the bones [11]. Ra223 was associated with a 30% lower risk of death in comparison to the control group of patients receiving the best standard of care, which resulted in a 3.6-month difference in median OS (14.9 months versus 11.3 months). This survival benefit was seen even though prostate-specific antigen (PSA) responses were rarely (16%) observed in patients undergoing Ra223 therapy.

Under numerous circumstances, PSA is a helpful prognostic biomarker in advanced prostate cancer [12]. As an example, a PSA decrease of ≥30% after 12 weeks of docetaxel chemotherapy (PSA30) is a surrogate marker of OS [13,14]. By contrast, in men treated with Ra223, neither PSA nor total alkaline phosphatase (ALP) or lactate dehydrogenase (LDH) declines at 12 weeks, are meeting surrogacy requirements for OS [15].

Early rises in PSA levels followed by declines are termed PSA flares. Such PSA surges have been observed in up to 30.6% of mCRPC patients treated with docetaxel or cabazitaxel chemotherapy but are more rarely associated with abiraterone or enzalutamide therapy [16,17,18,19,20,21,22,23]. It has been hypothesized that PSA flares are due to treatment-induced cancer cell lysis and PSA release into the bloodstream [16,24], increased differentiation of prostate cancer stem cells [16], or androgen receptor activation by corticosteroid co-medication [25]. Importantly, chemotherapy-associated PSA flares are not thought to negatively affect cancer-related outcomes [18,19].

Given a dearth of information on the role of PSA flares during Ra223 therapy [24,26,27,28], we sought to observe the prevalence and prognostic implications of PSA flares by analyzing the database of a retrospective cohort study of real-world Ra223 treatment at four cancer centres across Ontario/Canada [29]. We specifically compared the outcome between patients with PSA flares and those with or without PSA responses.

## 2. Materials and Methods

To study the PSA flare phenomenon in men with mCRPC treated with Ra223, we accessed a database established as part of a retrospective review of Ra223 utilization across four cancer centres in Ontario/Canada [29]. Briefly, all patients who received at least one dose of Ra223 via public funding from January 2015 to April 2016 in the four participating cancer centres were included. Of the 198 patients overall, we identified 134 subjects with a baseline and at least another two PSA readings obtained during the first 12 weeks of Ra223 therapy. Typically, PSA and other circulating markers, such as ALP, were obtained before each Ra223 administration. The PSA trends of those patients were used to assign each of them to one of three cohorts: (i) PSA flare: patients with an initial PSA increase of any degree followed by a subsequent decrease, either below baseline or below the peak on-treatment PSA level, but not below baseline; (ii) PSA response: immediate and continued decrease in PSA levels and ≥30% PSA decrease within 12 weeks; and (iii) PSA non-response: neither PSA flare nor PSA response.

We also studied numerous PSA- and ALP-based endpoints by applying definitions outlined in the ALSYMPCA protocol [11]. Namely, we analyzed the following PSA-based parameters: (i) PSA30 (≥30% reduction of PSA whenever compared to the baseline value), (ii) PSA50 (≥50% reduction of PSA whenever compared to the baseline value), and (iii) a confirmed PSA50 response (50% reduction of PSA whenever compared to the baseline value, confirmed by a second PSA value ≥ 4 weeks later). For ALP, we analyzed: (i) ALP30 (≥30% reduction of total ALP whenever compared to the baseline value); (ii) confirmed ALP30 response (≥30% reduction of total ALP whenever compared to baseline value, confirmed ≥ 4 weeks later); (iii) ALP50 (≥50% reduction of total ALP whenever compared to the baseline value); (iv) confirmed ALP50 response (≥50% reduction of total ALP whenever compared to baseline value, confirmed ≥4 weeks later); and (v) ALP normalization (return of ALP to value within the normal range (40–120 U/L) at 12 weeks, confirmed by two consecutive measurements ≥2 weeks apart, in patients with total ALP above upper limit of normal range (i.e., >120 U/L) at baseline.

Baseline patients’ characteristics, laboratory findings, and outcome measures were compared across the three cohorts by applying Chi-square, Fisher’s exact, or Kruskal–Wallis one-way variance tests for categorical or continuous variables, respectively. The Kaplan–Meier method and log-rank testing were used to evaluate and compare OS across patient cohorts. For the statistical analyses and preparation of graphs, we used the IBM SPSS^®^ (Statistical Package for the Social Sciences, Version 26) and GraphPad Prism^®^ (Version 5) platforms. A two-tailed *p*-value of <0.05 was considered statistically significant.

## 3. Results

### 3.1. Patient Demographics and Disease Characteristics

Overall, 27/134 (20.1%) patients experienced a PSA flare, of which 12/27 (44.4% of flare patients; 9.0% of all patients) had a PSA decrease below baseline levels after the initial PSA surge, while 15/27 (55.6% of flare patients; 11.2% of all patients) had a PSA decrease below the peak PSA level but not below baseline (Figure 1 and Figure 2a). Eleven (8.2%) patients were classified as PSA responders, whereas 96 (71.6%) men were considered PSA non-responders.

The median age of the entire study cohort was 75 years (IQR 67–80.8) (Table 1). The majority of patients were initially diagnosed with localized prostate cancer (61.2%), typically with a Gleason score of 8 or higher. Nearly 60% of patients underwent curative local treatment attempts: 27.6% had a prostatectomy alone, 23.1% underwent radiation alone, and 9% had a prostatectomy followed by radiation. One-third (32.1%) of patients received adjuvant androgen deprivation therapy (ADT). Bone was the most common site of metastases observed in 94% of patients, followed by nodal metastases in 14.9% of patients. Prior to Ra223, patients received a median of two lines of treatment for metastatic disease, excluding ADT. The vast majority of patients (95.5%) had been treated previously with abiraterone and/or enzalutamide, and 50.7% underwent prior docetaxel chemotherapy. About one quarter (27.6%) of all patients had experienced prior skeletal-related events, while 27/66 (41%) evaluable patients had initiated denosumab and/or zoledronic acid therapy before starting Ra223. The symptom burden as assessed by the Edmonton Symptom Assessment System (ESAS) Pain Sub-Score was found to be relatively low (i.e., a median of 2 out of a range from 0 to 10, with 0 representing the absence of pain). At a median follow-up of 9.5 months from the start of Ra223 therapy, 64.9% of patients were alive when the database was closed in June 2017. After treatment with Ra223, 26.8% of patients did not receive any additional therapy, while 25.8% went on to receive additional treatments such as docetaxel and enzalutamide. For the remaining 47.4% of patients, information on subsequent therapies was not available.

When comparing PSA Flare patients with PSA Responders and Non-Responders, the former had a higher prostatectomy rate, received fewer prior lines of systemic therapy, and had a higher exposure to antiresorptive agents (Table 1). However, the abiraterone/enzalutamide and docetaxel pre-treatment rates were similar across all three sub-cohorts. Similarly, there were no significant differences in the metastatic patterns across cohorts.

At the start of Ra223 therapy, the median PSA of all patients measured 75.5 μg/L (range: 0.83–4081; IQR: 27.6–162.3), the median ALP measured 109.0 U/L (range: 12–1633; IQR: 72.8–200), and the median hemoglobin (Hb) measured 120.5 g/L (range: 75–161; IQR: 111–132) (Table 2). While ALP and Hb values were similar across the sub-cohorts, baseline PSA values were significantly lower in PSA flare patients compared to PSA responders and non-responders (*p* = 0.0009).

### 3.2. Treatment Characteristics and Outcome Measures

Overall, patients received a median of 6 cycles of Ra223 (Table 3). A total of 70 (52.2%) patients received all six standard doses of Ra223; 24 (17.9%) received five doses; and 40 (29.9%) received four or fewer doses of treatment. Amongst sub-cohorts, the median number of Ra223 therapies was lower in PSA non-responders (i.e., 5, [range 2–6], compared to 6 [3–6] in PSA Flare and 6 [2–6] in PSA responders). Of PSA flare and PSA responders, 81.5% and 81.8%, respectively, received all Ra223 treatment cycles, whereas only 40.6% of PSA non-responders completed six cycles. The most common reason for treatment discontinuation was disease progression, with additional reasons including patient requests, low blood counts, and hospital admissions.

Among PSA flare patients, eighteen (66.7%) experienced a PSA flare after the first cycle of Ra223 treatment, six (22%) after the second cycle, and three (11%) after more than two cycles of Ra223.

PSA responders generally had superior PSA-based endpoint results compared to flare patients and non-responders (Table 3, Figure 2a). While—by definition—all PSA responders experienced a PSA30, only 22.2% of flare patients experienced such a decrease. About half of the PSA responders also experienced PSA50 and confirmed PSA50 responses, whereas the according rates for flare patients were 11.1% and 3.7%.

Compared to PSA non-responders, a higher proportion of responders and flare patients met ALP-based endpoints, namely ALP50 and confirmed ALP50 (Table 3). On the other hand, the PSA responder sub-cohort had the highest proportion of patients with abnormal ALP levels at baseline (45.5%), yet none of these patients reached a normalized ALP by week 12 of Ra223 treatment. The ALP normalization rate was 37.5% in flare patients and 5.4% in PSA non-responders. The rate of PSA non-responders with any degree of ALP decrease was 60.4%, and thus lower than in PSA responders (81.8%) and flare patients (81.5%) (Figure 2b).

The median OS from the start of Ra223 therapy for the whole study cohort was 16.8 months (95% CI: 13.4–20.2) (Table 3). There were no significant survival differences among the three sub-cohorts of patients (Figure 3).

## 4. Discussion

In this retrospective analysis of 134 patients undergoing Ra223 therapy, we found a PSA flare in 27 patients (20.1%), of whom 6 (22.2%) eventually achieved a PSA30 and 3 (11.1%) a PSA50. Despite the rarity of clinically meaningful PSA declines in PSA flare patients, the appearance of an initial PSA surge did not negatively affect OS when compared to PSA responders, who by definition all achieved a PSA30. The findings of the present study are comparable to those of men receiving taxane chemotherapy for mCRPC, for whom PSA flare rates ranging from 7.6% to 30.6% have been reported [16,17,18,19,20]. Moreover, the outcome of men with chemotherapy-induced PSA flares is similar to that of non-flare patients [18,19].

Up to now, Ra223-induced PSA flares have been described in three case reports and two retrospective series [24,26,27,28,30]. Modi et al. observed four out of 25 (16%) patients with a PSA response to Ra223 [28]. Among those four PSA responders, three experienced an initial PSA increase of maximally 4%, 117%, or 411% above baseline, respectively, followed by PSA declines first observed after two, four, and three cycles. Hyväkkä et al. performed a retrospective analysis of patients receiving Ra223 between 2014 and 2019 [30]. They observed PSA flares followed by PSA declines in 13 (8.1%) patients. Finally, Castello et al. analyzed Ra223-induced PSA flares in an overall cohort of 168 patients [24]. Compared to our series, they observed a higher PSA flare rate (35.7% versus 20.1%), yet a similar incidence of flare patients achieving a PSA decline below baseline (11.9% versus 9.0%). Furthermore, OS and progression-free survival were comparable in patients with a PSA flare versus patients with a PSA response without an initial surge. However, in contrast to our study, Castello et al. observed a significantly superior OS in PSA flare patients and PSA responders compared to PSA non-responders. Although the study by Castello et al. is similar to our analyses in several ways (e.g., number of study subjects, length of follow-up, and median baseline ALP), there are also differences. Their cohort was younger (median age 69 years versus 75 years), the median baseline PSA was lower (29.7 µg/L versus 75.5 µg/L), the Ra223 completion rate was higher (64.2% versus 52.2%), the frequency of PSA non-responders lower (35.1% versus 71.6%), and the median OS of both the entire cohort and of the PSA non-responders shorter (12.2 months versus 16.8 months and 11.5 months versus 17.8 months, respectively). On the other hand, the median OS of the PSA flare cohorts was similar between the two studies (18.3 months versus 16.8 months).

While PSA kinetics over the first 12 weeks of Ra223 therapy enabled the identification of three distinct response patterns as outlined in the present study, the latter were not associated with specific ALP profiles (Figure 2b). In fact, the majority of patients achieved a certain degree of ALP decline, even in the PSA non-responder cohort. Ra223 impacts the circulating levels of skeletal ALP either by directly targeting osteoblasts or indirectly by modifying the number and activity of prostate cancer cells [10]. Hence, Ra223-induced ALP declines are an expected, collateral consequence of Ra223 therapy, akin to neutrophil declines seen with many chemotherapeutics, but not clearly related to patient outcome [15]. The discordance between PSA and ALP responses is not fully understood. Since the alpha-emitting activity of Ra223 is limited to short distances and focused on areas of high bone turnover with high osteoblast density and activity, many prostate cancer cells might be beyond the reach of Ra223 [31,32].

While the present analysis provides further evidence that Ra223-associated PSA flares are not prognostic for poor outcomes, there are a number of limitations worth mentioning. First, although the data is representative of four cancer centres in Ontario/Canada, it was collected in a retrospective manner. Second, among the initial patient cohort of 198 patients [29], the database did not comprise sufficient PSA data regarding one-third of subjects. Third, only 52.2% of study subjects completed 6 cycles of Ra223, which is lower than in ALSYMPCA but comparable to other real-world series [33]. Finally, there is no universally accepted definition of PSA flares, and the nature of the database does not allow for a correlation of PSA flares with pain or bone scan flares.

## 5. Conclusions

In summary, a sizeable number of mCRPC patients undergoing Ra223 therapy experience a PSA flare. In the present study, the OS of such flare patients was comparable to men with immediate PSA declines or to PSA non-responders. ALP responses are commonly seen across all three patient cohorts, with distinct PSA response patterns. Further studies are needed regarding suitable biochemical surrogate markers of response to Ra223.

## Figures and Tables

**Figure 1 jcm-12-05604-f001:**
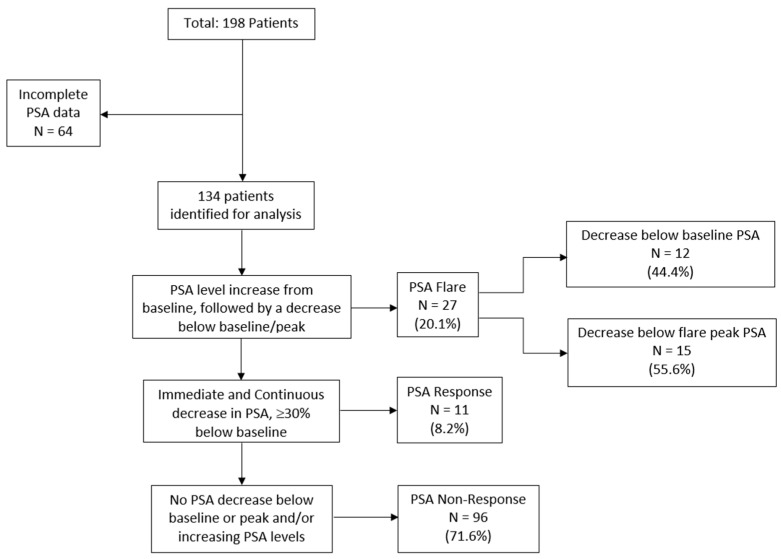
Flow Diagram: Depiction of patient identification and assignment to three distinct PSA response patterns, i.e., PSA flare, PSA response, and PSA non-response.

**Figure 2 jcm-12-05604-f002:**
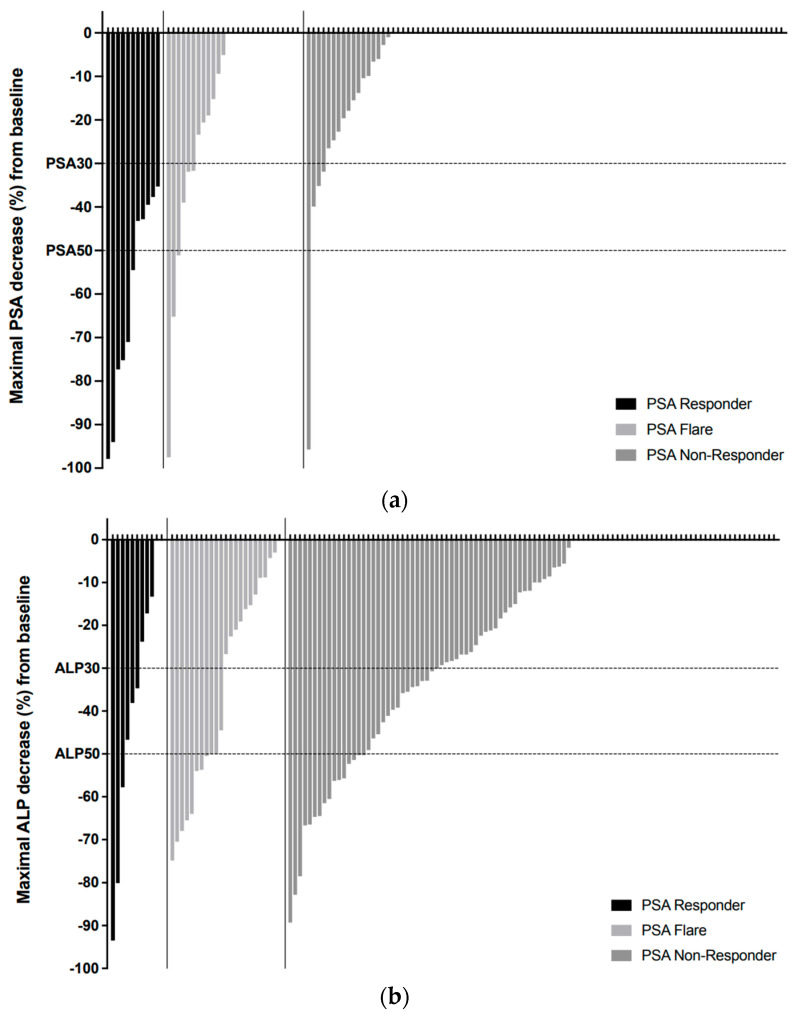
PSA and ALP Waterfall Plots: Maximal documented PSA ((**a**), **top panel**) or ALP ((**b**), **bottom panel**) decreases (expressed as percentual decline with respect to baseline values) during the first 12 weeks of Ra223 therapy, represented as waterfall plots. Patients without any degree of PSA or ALP decrease were coded as “0” and are represented with x-axis upward ticks only without downward bars.

**Figure 3 jcm-12-05604-f003:**
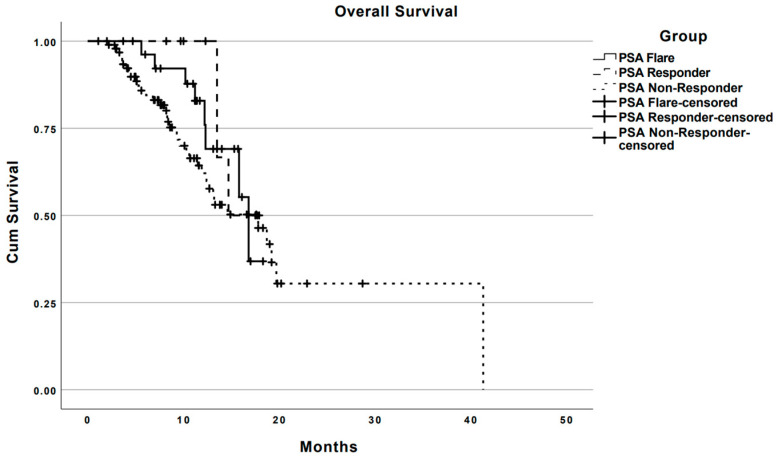
Overall Survival Analysis: The median overall survival of the entire study cohort was 16.8 months (95% CI: 13.4–20.2) and was found to be similar across subcohorts (*p* = 0.349): PSA flare 16.8 months (95% CI: 14.9–18.7); PSA responders 14.7 months (95% CI not applicable); and PSA non-responders 17.8 months (95% CI: 11.8–23.8).

**Table 1 jcm-12-05604-t001:** Patient characteristics.

	Total	PSA Flare	PSA Responders	PSA Non-Responders	*p*-Values
	N = 134	N = 27 (20.1%)	N = 11 (8.2%)	N = 96 (71.6%)	
**Age (years)**					0.3489
N	134	27	11	96	
Mean ± SD	73.8 ± 9.4	72.1 ± 10.4	76.8 ± 7.7	74.1 ± 9.2	
Range	55–93	55–92	65–91	56–93	
Median (IQR)	75.0 (67–80.8)	74.0 (64–78.5)	76.0 (72.5–79.5)	74.5 (67–81)	
**Initial stage**					0.4671
Localized	82 (61.2%)	14 (51.9%)	9 (81.8%)	59 (61.5%)	
Metastatic	18 (13.4%)	5 (18.5%)	0 (0.0%)	13 (13.5%)	
Unknown	34 (25.4%)	8 (29.6%)	2 (18.2%)	24 (25.0%)	
**Local therapy**					**0.007**
Prostatectomy	37 (27.6%)	13 (48.1%)	3 (27.3%)	21 (21.9%)	
Radiation	31 (23.1%)	5 (18.5%)	6 (54.5%)	20 (20.8%)	
Prostatectomy/Radiation	12 (9.0%)	0 (0.0%)	1 (9.1%)	11 (11.5%)	
No local therapy	54 (40.3%)	9 (3.33%)	1 (9.1%)	44 (45.8%)	
**Adjuvant androgen**					0.0647
**deprivation therapy**					
Yes	43 (32.1%)	8 (29.6%)	7 (63.6%)	28 (29.2%)	
No	91 (67.9%)	19 (70.4%)	4 (36.4%)	68 (70.8%)	
**Gleason Score**					0.7843
6	4 (3.0%)	0 (0.0%)	0 (0.0%)	4 (4.2%)	
7	40 (29.9%)	9 (33.3%)	3 (27.3%)	28 (29.2%)	
8–10	63 (47.0%)	12 (44.4%)	7 (63.6%)	44 (45.8%)	
Unknown	27 (20.1%)	6 (22.2%)	1 (9.1%)	20 (20.8%)	
**Bone metastases**					0.0964
Yes	126 (94.0%)	24 (88.9%)	11 (100.0%)	91 (94.8%)	
No	1 (0.7%)	0 (0.0%)	0 (0.0%)	1 (1.0%)	
Unknown	7 (5.2%)	3 (11.1%)	0 (0.0%)	4 (4.2%)	
**Nodal metastases**					0.3440
Yes	20(14.9%)	4 (14.8%)	0 (0.0%)	16 (16.7%)	
No	106 (79.1%)	20 (74.1%)	11 (100.0%)	75 (78.1%)	
Unknown	8 (6.0%)	3 (11.1%)	0 (0.0%)	5 (5.2%)	
**Visceral metastases**					0.4664
Yes	13 (9.7%)	3 (11.1%)	0 (0.0%)	10 (10.4%)	
No	113 (84.4%)	21 (77.8%)	11 (100.0%)	81 (84.4%)	
Unknown	8 (6.0%)	3 (11.1%)	0 (0.0%)	5 (5.2%)	
**Number of lines of prior therapies**					0.2634
Mean ± SD	2.0 ± 1.0	1.8 ± 0.9	2.4 ± 1.0	2.1 ± 1.1	
Median (range)	2 (0–5)	2 (1–4)	2 (1–4)	2 (0–5)	
Distribution					**0.0277**
0	2 (1.5%)	0 (0.0%)	0 (0.0%)	2 (2.1%)	
1	48 (35.8%)	12 (44.4%)	2 (18.2%)	34 (35.4%)	
2	37 (27.6%)	10 (37.0%)	5 (45.5%)	22 (22.9%)	
3	39 (29.1%)	3 (11.1%)	2 (18.2%)	34 (35.4%)	
4	5 (3.7%)	2 (7.4%)	2 (18.2%)	1 (1.0%)	
5	3 (2.2%)	0 (0.0%)	0 (0.0%)	3 (3.1%)	
**Prior Abiraterone and/or Enzalutamide**					0.7162
Yes	128 (95.5%)	26 (96.3%)	11 (100.0%)	91 (94.8%)	
No	6 (4.5%)	1 (3.7%)	0 (0.0%)	5 (5.2%)	
**Prior Docetaxel**					0.5978
Yes	68 (50.7%)	11 (40.7%)	6 (54.5%)	49 (51.0%)	
No	66 (49.3%)	16 (59.3%)	5 (45.5%)	47 (49.0%)	
**Prior Denosumab and/or Zoledronic Acid**					**0.0014**
Yes	27 (20.1%)	6 (22.2%)	0 (0.0%)	2 (2.1%)	
No	39 (29.1%)	5 (18.5%)	2 (18.2%)	32 (33.3%)	
Unknown	87 (64.9%)	16 (59.3%)	9 (81.8%)	62 (64.6%)	
**Prior skeletal related events**					0.7247
Yes	37 (27.6%)	7 (25.9%)	2 (18.2%)	28 (29.2%)	
No	97 (72.4%)	20 (74.1%)	9 (81.8%)	68 (70.8%)	
**ESAS Pain Score**					0.8731
N	75	19	7	49	
Mean ± SD	2.4 ± 2.6	2.4 ± 2.9	2.3 ± 1.8	2.5 ± 2.6	
Range	0–9	0–9	0–5	0–8	
Median (IQR)	2 (0–3.5)	2 (0–3)	2 (1–4)	2 (0–4)	
**Vital status June 2017**					0.6357
Alive	87 (64.9%)	19 (70.4%)	8 (72.7%)	60 (62.5%)	
Dead	47 (35.1%)	8 (29.6%)	3 (27.3%)	36 (37.5%)	
**Time to death/last follow up (months)**					0.1387
N	134	27	11	96	
Mean ± SD	9.8 ± 5.8	11.1 ± 3.8	12.1 ± 4.2	9.2 ± 6.3	
Range	1–40	3–18	4–17	1–40	
Median (IQR)	9.5 (5–13)	11 (10–15)	13 (9–17)	8 (4–12)	

IQR—interquartile range. The *p*-values of statistically significant findings across the three sub-cohorts (i.e., PSA flare, PSA responders, and PSA non-responders) are highlighted (bold print).

**Table 2 jcm-12-05604-t002:** Baseline laboratory findings.

	Total	PSA Flare	PSA Responders	PSA Non-Responders	*p*-Values
	N = 134	N = 27 (20.1%)	N = 11 (8.2%)	N = 96 (71.6%)	
**PSA (μg/L)**					**0.0009**
N	134	27	11	96	
Mean ± SD	195.5 ± 444.3	108.7 ± 341.2	563.1 ± 1255.0	177.7 ± 246.0	
Range	0.83–4081	0.91–1798	0.83–4081	1.34–1344	
Median (IQR)	75.5 (27.6–162.3)	31.8 (12.4–62.8)	83.9 (10.1–160)	95.2 (36.6–198.4)	
**ALP (U/L)**					0.8403
N	114	23	11	80	
Mean ± SD	183.1 ± 213.1	190.3 ± 227.5	196.6 ± 202.1	179.1 ± 212.8	
Range	12–1633	47–1020	45–739	12–1633	
Median (IQR)	109 (72.8–200)	99 (68–186)	105 (80–291)	113 (72.3–197.8)	
**Hb (g/L)**					0.1141
N	128	26	11	91	
Mean ± SD	120.1 ± 19.5	125.1 ± 17.9	116.2 ± 16.0	119.1 ± 20.2	
Range	75–161	75–149	85–139	12–161	
Median (IQR)	120.5 (111–132)	128.5 (116.8–137.8)	120.0 (108–125)	118.0 (110–132)	

IQR—interquartile range. The *p*-values of statistically significant findings across the three sub-cohorts (i.e., PSA flare, PSA responders, and PSA non-responders) are highlighted (bold print).

**Table 3 jcm-12-05604-t003:** Outcome measures.

	Total	PSA Flare	PSA Responders	PSA Non-Responders	*p*-Values
	N = 134	N = 27 (20.1%)	N = 11 (8.2%)	N = 96 (71.6%)	
**No. of Ra223 treatments**					**0.0001**
Mean ± SD	5.0 ± 1.3	5.7 ± 0.7	5.5 ± 1.2	4.7 ± 1.4	
Range	2–6	3–6	2–6	2–6	
**PSA30 ^1^**	21 (15.7%)	6 (22.2%)	11 (100%)	N/A	**<0.0001**
**PSA50 ^1^**	10 (7.5%)	3 (11.1%)	6 (54.5%)	N/A	**0.0090**
**Confirmed PSA50 ^1, 2^**	6 (4.5%)	1 (3.7%)	5 (45.5%)	N/A	**0.0041**
**ALP30**	42 (31.3%)	10 (37.0%)	5 (45.5%)	27 (28.1%)	0.3893
**Confirmed ALP30 ^2^**	32 (23.9%)	8 (29.6%)	5 (45.5%)	19 (19.8%)	0.1230
**ALP50**	21 (15.7%)	8 (29.6%)	3 (27.3%)	10 (10.4%)	**0.0286**
**Confirmed ALP50 ^2^**	14 (10.4%)	5 (18.5%)	3 (27.3%)	6 (6.3%)	**0.0300**
**Normalization of ALP**					**0.0170**
No. with abnormal ALP at baseline ^3^	50 (37.3%)	8 (29.6%)	5 (45.5%)	37 (38.5%)	
No. with ALP normalization ^4^	5 (10.0%)	3 (37.5%)	0 (0.00%)	2 (5.4%)	
**Overall Survival (months)**					0.349
Median (95% CI)	16.8 (13.4–20.2)	16.8 (14.9–18.7)	14.7 (N/A)	17.8 (11.8–23.8)	

^1^ From patients with a continuous decrease in PSA levels. ^2^ Confirmed values = confirmed 4 or more weeks later with a second value. ^3^ Percentage of patients with abnormal baseline ALP from the total cohort. ^4^ Percentage of patients with abnormal baseline ALP levels that normalized during Ra223 therapy. 95% CI—95% confidence interval. N/A—not applicable. The *p*-values of statistically significant findings across the three sub-cohorts (i.e., PSA flare, PSA responders, and PSA non-responders) are highlighted (bold print).

## Data Availability

The data presented in this study are available on request from the corresponding author. The data are not publicly available due to privacy restrictions.

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
