# Peer review of "Prevalence and Prognostic Implications of PSA Flares during Radium-223 Treatment among Men with Metastatic Castration Resistant Prostate Cancer"

_jcm, 2023, doi:10.3390/jcm12175604_

Round 1

Reviewer 1 Report

This is a very nice retrospective study including 134 PCa patients from 4 centers in Canada analyzing the significance of PSA-flare during the ra223 treatment to the patient outcome. The manuscript is generally well written, and the data is important for clinicians treating Pca patients. PSA responses are rarely observed in patients undergoing Ra223 therapy, and how to evaluate who benefits from ra223 remains challenging. Several rwe studies of ra223 already exist, but the focus on PSA flare phenomen here adds impact. Specific comments:

Please describe how patients from 4 centers were chosen for the cohort. Were all patients treated with ra223 in these centers included and during what years?

Please describe if patients received therapies after ra223 and what treatments did they receive.

It is unclear when the ALP and PSA values were measured, always before ra223 in each cycle?

What was the (real world) PFS for these patients – any differences between groups?

Flare does not seem to separate patients in terms of outcome. Others have reported OS differences between PSA responders and non-responders + suggest ALP as a biomarker of outcome. Flare group included both PSA responders and non-responders. Were there any differences in outcome measures and survival if flare patients were not separated but instead cohort was divided into PSA-responders and non-responders? The same with changes in ALP levels and their normalization – any differences in outcome in this cohort? This would give perspective to interpret the results regarding the flare group goving possibilities to see how the cohort compares to other studies.

There are significant p-values in bold in tables, but it is not always clear between what groups the significance was observed - please check that all is described clearly in text.

46. typo: [5. -> [5].

61. typo: [10. -> [10].

68: typo: [13, -> [13],

71: typo: [15. -> [15].

83: Please define PSA flare – how much should the value increase during the treatment from baseline?

128: It is not clear where the ticks are in the figure?

131: sixty percent -> 60 %

156: This sentence is not clear. What is the p-value describing?

170: Any other reasons to discontinue treatment? These should be clarified.

Table 3: #-> number or no.

214: typo: [15. -> [15].

215: typo: [21. -> [21].

215: Some references are missing here after the first sentence, please refer to the case reports + retrospective data, only one ref 21 is added now. Other studies have also analyzed retrospectively the significance of PSA responses to outcome and PSA flare, eg. Hyväkkä A et al Cancer Medicine 2022 found similar to Castello et al differences in OS between PSA responders and non-responders and reported flare rate of approximately 8 %.

216: [21. -> [21].

224: your study -> this/our study?

Author Response

Reviewer #1:

This is a very nice retrospective study including 134 PCa patients from 4 centers in Canada analyzing the significance of PSA-flare during the ra223 treatment to the patient outcome. The manuscript is generally well written, and the data is important for clinicians treating Pca patients. PSA responses are rarely observed in patients undergoing Ra223 therapy, and how to evaluate who benefits from ra223 remains challenging. Several rwe studies of ra223 already exist, but the focus on PSA flare phenomen here adds impact. Specific comments:

  • We would like to thank reviewer #1 for the positive feed-back and for raising the issues addressed below. Addressing the comments certainly improved the manuscript.

Please describe how patients from 4 centers were chosen for the cohort. Were all patients treated with ra223 in these centers included and during what years?

  • The requested information has been added (lines 84-86).

Please describe if patients received therapies after ra223 and what treatments did they receive.

  • The requested information has been added (lines 153-156).

It is unclear when the ALP and PSA values were measured, always before ra223 in each cycle?

  • Typically, PSA, ALP and other circulating markers were obtained before each Ra223 administration (lines 88-89).

What was the (real world) PFS for these patients – any differences between groups?

  • The retrospective study design focused on overall survival and early treatment discontinuation as endpoints. Hence, PFS information was not collected, and the database was locked in June 2017.

Flare does not seem to separate patients in terms of outcome. Others have reported OS differences between PSA responders and non-responders + suggest ALP as a biomarker of outcome. Flare group included both PSA responders and non-responders. Were there any differences in outcome measures and survival if flare patients were not separated but instead cohort was divided into PSA-responders and non-responders? The same with changes in ALP levels and their normalization – any differences in outcome in this cohort? This would give perspective to interpret the results regarding the flare group goving possibilities to see how the cohort compares to other studies.

  • In the main analysis of our dataset (Cheng et al. Cancer Manag Res. 2019 Oct 31;11:9307-9319; entire study cohort = 198 patients), neither conventional PSA parameters (such as PSA50) nor ALP-based parameters were associated with OS. Moreover, the present manuscript represents sub-cohort analyses. Hence, we decided not to pursue statistical testing of such parameters in the present manuscript.

There are significant p-values in bold in tables, but it is not always clear between what groups the significance was observed - please check that all is described clearly in text.

  • A clarifying statement was added at the bottom of each table.
  1. typo: [5. -> [5].
  2. typo: [10. -> [10].

71: typo: [13, -> [13],

74: typo: [15. -> [15].

  • Thanks for pointing this out; the revised manuscript contains proper formatting.

89: Please define PSA flare – how much should the value increase during the treatment from baseline?

  • There is no universally accepted definition of PSA flare; in the revised manuscript the definition of PSA flare used for our analysis has been clarified (line 91).

134: It is not clear where the ticks are in the figure?

  • Thanks for bringing this to our attention; the revised Figure legend has been modified.

138: Sixty percent -> 60 %

  • ‘Sixty’ was changed to ‘60’ (line 139)

168: This sentence is not clear. What is the p-value describing?

  • Thanks for bringing this to our attention; this paragraph has been rephrased during the revision of the manuscript (lines 169-174).

188: Any other reasons to discontinue treatment? These should be clarified.

  • The according sentence was rephrased as follows: The most common reason for treatment discontinuation was disease progression, with additional reasons including patient request, low blood counts, and hospital admissions. (lines 187-189)

Table 3: #-> number or no.

  • Thanks for pointing this out; # was replaced by No.

237: typo: [15. -> [15].

239: typo: [21. -> [21].

  • Thanks for pointing this out; the revised manuscript contains proper formatting.

239: Some references are missing here after the first sentence, please refer to the case reports + retrospective data, only one ref 21 is added now. Other studies have also analyzed retrospectively the significance of PSA responses to outcome and PSA flare, eg. Hyväkkä A et al Cancer Medicine 2022 found similar to Castello et al differences in OS between PSA responders and non-responders and reported flare rate of approximately 8 %.

  • Thanks for bringing Hyväkkä A et al Cancer Medicine 2022 to our attention; this reference has been integrated in the revised Discussion (lines 241-242). The missing references have been added (lines 236-237).

239: [21. -> [21].

  • Thanks for bringing this to our attention; the revised manuscript contains the proper formatting.

250: your study -> this/our study?

  • Thanks for pointing this out; we have corrected this typo during the revision of the manuscript (line 248).

Reviewer 2 Report

The authors should be congratulated for their work and for addressing an important topic. Only a few points warrant mentions:

Minor comments:

1.    In the “Results” section, the authors are invited to report also inter-quarter range along with medians in the text. Moreover, reporting results, the authors are invited to write statistically significant p-values in the text to give a more scientifical sound.

2.    In the “Results” section, line 156, it is not clear which parameter is related to the reported p-value.

3.    In the “Discussion” section, line 229, the median age of the involved men is different from the one reported in the “Results” section. I suggest the authors check their analysis's consistency and verify the reported data.

4.    I invite the authors to report more information about the epidemiology of the disease and about the difficulties in its management due to the lack of predictive and prognostic tools as much in metastatic hormone-sensitive tumors as in the castration-resistant ones, just as has been done in these manuscripts that need to be taken in account: https://doi.org/10.1016/j.critrevonc.2020.102992https://doi.org/10.1159%2F000509434

minor editing.

Author Response

Reviewer #2:

The authors should be congratulated for their work and for addressing an important topic.

  • Thanks for the positive review and the insightful comments that helped to improve the manuscript.

Only a few points warrant mentions:

Minor comments:

In the “Results” section, the authors are invited to report also inter-quarter range along with medians in the text. Moreover, reporting results, the authors are invited to write statistically significant p-values in the text to give a more scientifical sound.

  • We have revised the manuscript accordingly.
  1. In the “Results” section, line 168, it is not clear which parameter is related to the reported p-value.
  • Thanks for bringing this to our attention; this sentence has been rephrased during the revision of the manuscript (lines 169-174).
  1. In the “Discussion” section, line 254, the median age of the involved men is different from the one reported in the “Results” section. I suggest the authors check their analysis's consistency and verify the reported data.
  • Thanks for bringing our attention to this mistake that was rectified during the revisions (line 252).
  1. I invite the authors to report more information about the epidemiology of the disease and about the difficulties in its management due to the lack of predictive and prognostic tools as much in metastatic hormone-sensitive tumors as in the castration-resistant ones, just as has been done in these manuscripts that need to be taken in account: https://doi.org/10.1016/j.critrevonc.2020.102992, https://doi.org/10.1159%2F000509434. 
  • We have modified the Introduction accordingly, including integrating reference https://doi.org/10.1016/j.critrevonc.2020.102992. However, https://doi.org/10.1159%2F000509434 “Clinical Characteristics of Metastatic Prostate Cancer Patients Infected with COVID-19 in South Italy” seems vaguely related to the topic of our manuscript and instead we have added the following two references:
    • Hawley et al. Oncologist. 2023 Feb 8;28(2):93-104.
    • Sartor O. Clin Cancer Res. 2023 Aug 1;29(15):2745-2747

(references added = 6-8 in reference list, inserted in text = lines 43+44) 

Round 2

Reviewer 2 Report

Authors answered all comments and suggestions.

Minor editing